# The Multi-Kinase Inhibitor EC-70124 Is a Promising Candidate for the Treatment of FLT3-ITD-Positive Acute Myeloid Leukemia

**DOI:** 10.3390/cancers14061593

**Published:** 2022-03-21

**Authors:** Belen Lopez-Millan, Paula Costales, Francisco Gutiérrez-Agüera, Rafael Díaz de la Guardia, Heleia Roca-Ho, Meritxell Vinyoles, Alba Rubio-Gayarre, Rémi Safi, Julio Castaño, Paola Alejandra Romecín, Manuel Ramírez-Orellana, Eduardo Anguita, Irmela Jeremias, Lurdes Zamora, Juan Carlos Rodríguez-Manzaneque, Clara Bueno, Francisco Morís, Pablo Menendez

**Affiliations:** 1Department of Biomedicine, Josep Carreras Leukemia Research Institute, School of Medicine, University of Barcelona, 08036 Barcelona, Spain; fgutierrez@carrerasresearch.org (F.G.-A.); rafael.diaz@genyo.es (R.D.d.l.G.); heleia.roca@gmail.com (H.R.-H.); mvinyoles@carrerasresearch.org (M.V.); arubio@carrerasresearch.org (A.R.-G.); rsafi@carrerasresearch.org (R.S.); jcastano@bst.cat (J.C.); promecin@carrerasresearch.org (P.A.R.); lzamora@carrerasresearch.org (L.Z.); cbueno@carrerasresearch.org (C.B.); 2GENYO, Centre for Genomics and Oncological Research, Pfizer, Universidad de Granada, Junta de Andalucía, 18016 Granada, Spain; juancarlos.rodriguez@genyo.es; 3Red Española de Terapias Avanzadas (TERAV), Instituto de Salud Carlos III (ISCIII), 28029 Madrid, Spain; mramirezo.hnjs@salud.madrid.org; 4EntreChemSL, 33011 Oviedo, Spain; paula.costales@artiospharma.com (P.C.); fmv@entrechem.com (F.M.); 5Department of Pediatric Hematology and Oncology, Hospital Infantil Universitario Niño Jesús, Instituto de Investigación Sanitaria La Princesa, 28006 Madrid, Spain; 6Servicio de Hematología, Hospital Clínico San Carlos, IdISSC, School of Medicine, Universidad Complutense de Madrid, 28040 Madrid, Spain; eanguita@med.ucm.es; 7Research Unit Apoptosis in Hematopoietic Stem Cells, Helmholtz Zentrum München, 85764 Munich, Germany; irmela.jeremias@helmholtz-muenchen.de; 8Hematology Department, ICO-Hospital Germans Trias i Pujol, 08916 Barcelona, Spain; 9Centro de Investigación Biomédica en Red–Oncología (CIBERONC), 28029 Madrid, Spain; 10Instituciò Catalana de Recerca i Estudis Avançats (ICREA), 08010 Barcelona, Spain

**Keywords:** AML, EC-70124 multi-kinase inhibitor, FLT3-ITD mutation, FLT3 inhibitor, AML preclinical model

## Abstract

**Simple Summary:**

Patients with AML harboring constitutively active mutations in the FLT3 receptor generally have a poor prognosis (FLT3-ITD^MUT^). Despite the fact that several FLT3 inhibitors have been developed, clinical responses are commonly partial or not durable, highlighting the need for new molecules targeting FLT3-ITD^MUT^. Here, we tested EC-70124, a hybrid indolocarbazole analog from the same chemical space as midostaurin (a well-known FLT3 inhibitor). Our in vitro and in vivo experiments showed that EC-70124 exerts a robust and specific antileukemia activity against FLT3-ITD^MUT^ AML cells while sparing healthy hematopoietic cells. Collectively, EC-70124 is a promising and safe agent for the treatment of this aggressive type of AML.

**Abstract:**

Acute myeloid leukemia (AML) is the most common acute leukemia in adults. Patients with AML harboring a constitutively active internal tandem duplication mutation (ITD^MUT^) in the FMS-like kinase tyrosine kinase (FLT3) receptor generally have a poor prognosis. Several tyrosine kinase/FLT3 inhibitors have been developed and tested clinically, but very few (midostaurin and gilteritinib) have thus far been FDA/EMA-approved for patients with newly diagnosed or relapse/refractory FLT3-ITD^MUT^ AML. Disappointingly, clinical responses are commonly partial or not durable, highlighting the need for new molecules targeting FLT3-ITD^MUT^ AML. Here, we tested EC-70124, a hybrid indolocarbazole analog from the same chemical space as midostaurin with a potent and selective inhibitory effect on FLT3. In vitro, EC-70124 exerted a robust and specific antileukemia activity against FLT3-ITD^MUT^ AML primary cells and cell lines with respect to cytotoxicity, CFU capacity, apoptosis and cell cycle while sparing healthy hematopoietic (stem/progenitor) cells. We also analyzed its efficacy in vivo as monotherapy using two different xenograft models: an aggressive and systemic model based on MOLM-13 cells and a patient-derived xenograft model. Orally disposable EC-70124 exerted a potent inhibitory effect on the growth of FLT3-ITD^MUT^ AML cells, delaying disease progression and debulking the leukemia. Collectively, our findings show that EC-70124 is a promising and safe agent for the treatment of AML with FLT3-ITD^MUT^.

## 1. Introduction

Acute myeloid leukemia (AML), the most common acute leukemia in adults, is a clinically and biologically heterogeneous disease characterized by the rapid proliferation and accumulation of immature myeloid blasts in bone marrow (BM) [1,2,3]. Internal tandem duplication (ITD) or kinase domain mutations of the Fms-like tyrosine kinase 3 (FLT3) gene represent genetic alterations frequently found in AML (~a third of patients) [4]. These mutations constitutively activate the tyrosine kinase receptor FLT3 and promote cell growth, survival and antiapoptotic signaling via several downstream targets including PI3K/AKT/mTOR and STAT5 [5,6]. FLT3-ITD mutations (FLT3-ITD^MUT^) represent approximately 80% of all FLT3 mutations and are associated with a high rate of relapse and poor clinical outcome [1,7,8,9]. As a consequence, there have been increasing efforts to develop potent targeted inhibitors of the FLT3 tyrosine kinase [10]. Despite initial challenges with the first-generation tyrosine kinase inhibitors (TKIs), which were associated with systemic toxicities, off-target effects and suboptimal target inhibition, there has been recent clinical success, with some TKIs including midostaurin and gilteritinib being clinically approved by the USA Food and Drug Administration (FDA) and the European Medicines Agency (EMA) as either monotherapy or combined with conventional chemotherapy for patients with de novo or relapse/refractory FLT3-ITD^MUT^ AML [11,12,13,14]. Unfortunately, however, most clinical responses to such TKIs are either partial or transient, and many patients ultimately relapse with the common acquisition of secondary mutations in FLT3 or the upregulation of compensatory oncogenic pathways [15,16]. In addition, midostaurin is currently administered in combination with cytarabine and daunorubicin, thus preventing its administration in elderly patients and/or in those with comorbidities [12]. Accordingly, there is a clinical need for the development of new FLT3 inhibitors to broaden the range of therapeutic arsenal for patients with FLT3-TD^MUT^ AML.

EC-70124 is a hybrid indolocarbazole analog from the same chemical space as midostaurin which has a potent multi-kinase inhibitory spectrum that impacts key signaling kinases implicated in pro-survival and proliferative cellular pathways [17]. The antitumor activity of EC-70124 has been demonstrated in solid tumors including sarcoma, glioblastoma, colorectal, prostate and breast cancers, associated with the inhibition of a wide variety of targets including components of PI3K/AKT/mTOR or JAK/STAT pathways [18,19,20,21,22,23]. In addition, EC-70124 was recently demonstrated to exert a potent and selective inhibitory effect on AML-related kinases such as FLT3 that is even higher than that of midostaurin [21]. Here, we sought to further characterize the antileukemia activity of EC-70124 as monotherapy against FLT3-ITD^MUT^ AML primary cells in both in vitro and in vivo models. We show that EC-70124 has a potent and selective cytotoxic effect on FLT3-ITD^MUT^ cells while sparing FLT3-ITD^WT^ cells and healthy hematopoietic (stem/progenitor) cells (HSPC). Our results support EC-70124 as a novel and safe TKI in FLT3-ITD^MUT^.

## 2. Materials and Methods

### 2.1. AML Cell Lines and Primary AML and Healthy HSPC

The FLT3-ITD^MUT^ AML cell line MV4;11 was obtained from the DSMZ (Braunschweig, Germany). The FLT3-ITD^WT^ cell line HL60 was kindly provided by Prof. Luciano Di Croce (CRG, Barcelona, Spain). The MOLM-13-Luc+ (MOML-13) and Baf3 cell lines were kindly provided by Prof. Michael Andreeff (MD Anderson Cancer Center, Houston, TX, USA). All cell lines were cultured in RPMI medium supplemented with fetal bovine serum and antibiotics (Gibco, Invitrogen, Paisley, UK). A luciferase-expressing FLT3-ITD^MUT^ AML patient-derived xenograft (PDX) (FLT3-ITD^MUT^-Luc+ AML-640 PDX) was generated and characterized as previously described by Jeremias’ Laboratory (Helmholtz Zentrum, München, Germany) [24,25]. Primary AML cells were obtained from fresh bone marrow (BM) aspirates from the Hospital Clínico San Carlos and the Hospital Infantil Niño Jesús (Madrid, Spain). BM-derived leukemic cells were isolated using Ficoll-Paque Plus (GE Healthcare) density gradient centrifugation. Table 1 summarizes the main clinical/cytogenetic/molecular characteristics of the patients’ cells. Fresh peripheral blood (PB) and cord blood (CB) units were obtained from healthy donors from the Catalonia Blood Tissue Bank following the institutional guidelines approved by our local Institutional Review Board. PB- and CB-derived mononuclear cells were also isolated using Ficoll-Paque density gradient centrifugation. CB-derived CD34+ HSPCs were magnetic-activated cell sorting (MACS)-purified using the human CD34 MicroBead kit and the AutoMACS device (Miltenyi Biotec, Madrid, Spain), as previously reported [26,27,28]. The purity of the CD34+ fraction was consistently >95%. Primary cells were cultured in StemSpan medium (Stem Cell Technologies, Vancouver, BC, Canada) supplemented with the hematopoietic cytokines Stem Cell Factor (100 ng/mL), FLT3 ligand (100 ng/mL), IL3 (10 ng/mL, all from PeproTech, London, UK) and antibiotics (Gibco). Cultures were maintained in a humidified atmosphere with 5% CO_2_ at 37 °C. The study was approved by the Institutional Review Board of the Clinic Hospital of Barcelona (HCB/2014/0687), and samples were accessed upon signed informed consent.

### 2.2. Drugs

EC-70124 was synthesized by EntreChem S.L. (Oviedo, Spain) using a proprietary process [21]. Midostaurin (PKC-412) was synthesized from staurosporine (Biomar Microbial Technologies, León, Spain), and the identity of the isolated product was verified by comparison with an authentic sample using high-performance liquid chromatography and nuclear magnetic resonance [21]. EC-70124 and midostaurin were reconstituted in DMSO and stored at −20 °C.

### 2.3. Cytotoxicity and Apoptosis Assays

The cytotoxic effect of the drugs on AML cell lines and Baf3 cells was assessed using the Cell Counting Kit-8 (which is based on reduction of the sulfonated tetrazolium salt WST-8) (Sigma-Aldrich, St. Lous, MO, USA) [28] and MTT assay, respectively, as described [28]. Briefly, 5000 AML cells or 60,000 Baf3 cells were plated in a 96-well plate and incubated with increasing concentrations of the corresponding drug for 48 and 96 h, respectively. Primary samples were incubated with EC-70124 (or midostaurin) at 200 nM (~IC50 average of all cell lines assayed in Puente-Moncada et al. [21]) for 48 h, and cell viability was measured using the Annexin-V/7-AAD Apoptosis Detection Kit (BD Biosciences, Madrid, Spain) on a FACS Canto-II cytometer, as described in [29]. Myeloid, B cell and T cell populations from HD-PBMCs were immunophenotyped by flow cytometry using the monoclonal antibodies CD33-APC, CD19-BV421 and CD3-PE, respectively.

### 2.4. Colony-Forming Unit (CFU) and Cell Cycle Assays

Cells were treated with 200 nM EC-70124 overnight (CB-CD34+ and primary AML cells) or for 48 h (AML cell lines). For the CFU formation assay, a total of 5 × 104 primary AML cells, 5 × 102 CB-CD34+ cells or 1 × 103 AML cells (cell lines) were plated onto methylcellulose H4434 medium (StemCell Technologies, Saint Égrève, France). Colonies were counted and scored at day 14 (CB-CD34+ and primary AML cells) or day 7 (AML cell lines) using standard morphologic criteria, as described in [27,29,30,31]. For cell cycle analysis, cells were fixed with 70% ice-cold ethanol after treatments. DNA was stained with a propidium iodide (PI) staining solution and cells were analyzed by flow cytometry (FACSCanto II, Becton Dickinson, Madrid, Spain), as described in [32]. Data were evaluated using ModFit (VeritySoftware).

### 2.5. Western Blot Analysis

A total of 10^7^ MOLM-13 cells were treated with the different drugs at the indicated concentrations for 4 h. Cells were then lysed in 1% Triton X-100 lysis buffer supplemented with a complete protease inhibitor cocktail (Roche, Diagnostic, Basel, Switzerland) and phosphatase inhibitors (Sigma-Aldrich, St. Lous, MO, USA), as detailed in [33]. Proteins were separated by SDS-PAGE and transferred to polyvinylidene difluoride membranes (Amersham Bioscience, Merck KGaA, Darmstadt, Alemania). Primary antibodies (Cell Signaling Technology, Danvers, MA, USA) were used against the phosphorylated (p) and total forms of Stat5 and Akt (1:1000), S6, BAD, Erk and Akt (1:2000 and 1:1000, for phosphorylated and total forms, respectively). For signaling detection, appropriate secondary (anti-rabbit IgG peroxidase-conjugated and anti-mouse IgG peroxidase 1:3000, Calbiochem) antibodies were used and the reaction was visualized using the Bio-Rad Chemidoc MP Imaging system.

### 2.6. NSG Mice Xenotransplantation and Engraftment Analysis

All experimental procedures were approved by the Animal Care Committee of The Barcelona Biomedical Research Park (HRH-17-0007). We used 8–14-week-old NOD.Cg-PrkdcscidIl2rgtm1Wjl/SzJ (NSG) mice housed under pathogen-free conditions. EC-70124 was dosed daily by oral gavage at 20 mg/kg [21] and tumor burden was monitored at the indicated time points by bioluminescence using the Xenogen in vivo imaging system (IVIS, Perkin Elmer). For the MOLM-13 model, 103 cells (n = 10 mice) were intra-BM (IBM) transplanted into non-irradiated NSG mice [34,35]. On the next day, mice were randomized into treatment groups and were treated for 15 days. At day 20, mice were sacrificed and cells from the BM and PB were stained with anti-human HLA-ABC-FITC, CD33-PE and CD45-APC antibodies (BD Biosciences, Madrid, Spain) to analyze leukemic engraftment by flow cytometry. For the FLT3-ITD^MUT^-AML PDX model, 105 cells were transplanted intravenously (i.v.) into sublethally (2 Gy) irradiated mice (n = 14 mice), as described in [36]. Engraftment was monitored weekly by IVIS until it was detectable in most mice (week 7). At this time, BM and PB were collected by aspiration and facial vein bleeding, respectively, and human grafts were immunophenotyped by flow cytometry using HLA-ABC-PE, CD45-FITC and CD33-APC antibodies. Mice were homogeneously divided into the treatment groups and were treated for 18 days. BM and PB were also collected at the end of the treatment and mice were left untreated for 65 days to follow-up potential relapse by IVIS. Mice were sacrificed when disease symptoms were evident and/or IVIS signal was incompatible with life.

### 2.7. Targeted Next Generation Sequencing (NGSeq)

A targeted NGSeq study was performed on ex vivo FACS-sorted AML cells retrieved from four mice (AML-640 PDX), treated (two mice) or untreated with EC-70124 (two mice) at the endpoint. The NGS panel included 76 genes related to myeloid disorders. Briefly, DNA was extracted from the AML cells and the libraries were prepared using the Haematology OncoKitDx (Imegen–Healthincode group^®^). Target regions were then captured and amplified by hybridizing the DNA library with the above-mentioned kit (Imegen, Healthincode^®^). The captured and amplified DNA was sequenced using NextSeq550 platform (Illumina^®^). The results were analyzed using DataGenomics platform as described elsewhere [37,38,39].

### 2.8. Statistical Analyses

Data are expressed as mean ± SEM of different independent experiments unless otherwise specified. Statistical comparisons were performed using unpaired Student’s t test, except for comparison between treatments for the same patient (paired Student’s test). Differences between treatments along the days in PDX model were performed using multiple t test. Statistical analyses, normality test (Shapiro–Wilk) and IC50 were performed using GraphPad Prism v6.0 (GraphPad Software Inc. San Diego, CA, USA). Statistical significance was defined as a *p*-value < 0.05.

## 3. Results

### 3.1. EC-70124 Has a Potent Cytotoxic Effect on FLT3-ITD^MUT^ AML Cells In Vitro

EC-70124 is an indolocarbazole from the same chemical space as midostaurin, with a recently demonstrated inhibitory activity against AML-related kinases and a greater potency than midostaurin for wild-type FLT3 and FLT3-ITD [21]. As these latter findings were based on an assay platform evaluating the dissociation constants of both compounds for selected kinases and disease relevant mutants, we were prompted to further functionally evaluate and compare the efficacy of EC-70124 and midostaurin against FLT3-ITD^MUT^ AML cells. Cytotoxicity assays based on tetrazolium dye reduction (WST-8) revealed a comparable efficacy for both inhibitors, with similar levels of half-maximal inhibitory concentration (IC50) in FLT3-ITD^MUT^ AML cells (17 nM and 14 nM in MOLM-13 and MV4;11 cells, respectively, for EC-70124; and 8 nM and 13 nM in MOLM-13 and MV4;11 cells, respectively, for midostaurin) (Figure 1a). EC-70124 treatment of primary FLT3-ITD^MUT^ AML patient cells had a stronger effect on cell viability than midostaurin (both 200 nM, during 48 h, Figure 1b). The inhibitory activity of EC-70124 was also assessed by CFU assays in FLT3-ITD^MUT^ AML patient cells and cell lines (Figure 1b,c). Up to a 50% decrease in the clonogenic potential after overnight pre-treatment with EC-70124 was observed with primary AML cells (17 ± 5 vs. 9 ± 3 CFUs in untreated and EC-70124-treated, respectively) (Figure 1b, right panel). This inhibitory activity was five- to six-fold more pronounced in both MOLM-13 and MV4;11 FLT3-ITD^MUT^ AML cell lines (Figure 1c). Mechanistically, and very similar to midostaurin, EC-70124-mediated cytotoxicity was associated with an induction of apoptosis and severe cell cycle arrest through specific inhibition (dephosphorylation) of p-ERK and STAT5, one of the most important downstream targets in constitutively-activated FLT3 [40] (Figure 1d–f). Although EC-70124 induced a mild decrease in pAKT, we detected a strong inhibition of master downstream components including pS6 and pBAD (Figure 1f). Lysates from cells treated with cytarabine and idarubicin (standard-of-care treatment in AML) were included as controls to confirm the specificity of EC-70124 on FLT3 signaling. Indeed, cytarabine and idarubicin increased the phosphorylation of most components of the FLT3 signaling, likely reflecting a damage-sensing cellular response (Figure 1f). Taken together, these data indicate that EC-70124 exerts a potent and specific cytotoxic effect on FLT3-ITD^MUT^ AML cells through inhibition of FLT3 downstream signaling.

### 3.2. EC-70124 Has No Evident Effect on FLT3-ITD^WT^ AML Cells and Spares Healthy HSPCs and Mature Blood Cells

We next tested the specificity of EC-70124 by assaying its effects on FLT3-ITD^WT^ AML cells using both FLT3-ITD^WT^ AML primary cells and the cell lines HL60 and THP1. Cytotoxicity analysis revealed that the EC-70124 IC50 was considerably higher in FLT3-ITD^WT^ cells than in FLT3-ITD^MUT^ cells (72 µM (HL60) and 77 nM (THP1) vs. 17 nM (MOLM-13) and 14 nM (MV4:11)) (Figure 1g). While EC-70124 treatment caused a decrease in cell viability of primary FLT3-ITD^WT^ AML patient cells (Figure 1h), the effect was much less pronounced than that observed in FLT3-ITD^MUT^ cells (Figure 1b). Treatment with EC-70124 did not affect the clonogenic potential (CFUs) of either FLT3-ITD^WT^ AML patient cells (Figure 1h) or FLT3-ITD^WT^ AML cell lines (Figure 1i), Accordingly, EC-70124 had no effect on apoptosis or cell cycle progression in FLT3-ITD^WT^ cells (Figure 1j,k). Finally, we performed a cytotoxicity assay on Baf3-WT or FLT3-ITD-overexpressing BaF cells (Baf3-ITD) treated with increasing concentrations of EC-70124 for 96 h (Figure 1l). The IC50 of EC-70124 was way lower in Baf3-ITD cells than in Baf3-WT cells (0.2 mM vs. 180 mM), confirming the specificity of EC-70124 for FLT3-ITD^MUT^ AML cells.

We next evaluated the cytotoxicity of EC-70124 on healthy mature B cells, myeloid cells and T cells (Figure 2a) and on CD34+ HSPCs (Figure 2b) by treating cells overnight with 200 nM EC-70124. EC-70124 revealed no cytotoxicity on either healthy mature circulating blood cells or HSPCs (Figure 2a,b). EC-70124 did not affect either the clonogenic capacity of the healthy CD34+ HSPCs (Figure 2b, right panel), demonstrating that EC-70124 has no detectable cytotoxic effects on normal hematopoiesis.

### 3.3. EC-70124 Exhibits Antileukemic Activity in FLT3-ITD^MUT^ AML Cells In Vivo

The oral availability and pharmacodynamics of EC-70124 were recently evaluated in a subcutaneous model of AML [21]. To extend and expand these observations, we tested its efficacy in two preclinical models of AML: an aggressive and systemic model using MOLM-13 cells and a robust FLT3-ITD^MUT^-Luc+ PDX model. For the MOLM-13 model, 1000 cells were IBM-transplanted into non-irradiated NSG mice, and due to the aggressiveness of this model, mice were randomized into groups and treatment was started the next day. Mice were dosed orally once daily with either vehicle or EC-70124 (20 mg/kg) for 15 days and were monitored by IVIS once weekly (Figure 3a). Mice were sacrificed at day 20 and engraftment levels were analyzed in BM and PB by flow cytometry. During the treatment period, EC-70124-treated mice showed significantly lower signals of leukemia than mice vehicle-treated mice, as measured by IVIS. At the endpoint, mice treated with EC-70124 had a significantly lower AML burden in both BM and PB than mice dosed with vehicle (Figure 3b). Next, we tested the antileukemic activity of EC-70124 using a PDX model with an FLT3-ITD^MUT^-Luc+ AML sample. A total of 100,000 PDX cells were i.v. injected into irradiated NSG mice and engraftment was monitored by IVIS weekly until mice showed a detectable signal. At this time, mice were randomized into two groups and were dosed orally once daily for 15 days. BM samples were taken by aspirate at the beginning and at the end of the treatment, and graft levels were analyzed by flow cytometry (Figure 3c). Unlike control mice, EC-70124-treated mice displayed the same low levels of engraftment at the beginning and at the end of the treatment, as monitored by IVIS and flow cytometry (Figure 3c,d). Quantification of the IVIS signal revealed that mice dosed with EC-70124 had a significantly lower leukemia burden than vehicle-treated mice at the end of the treatment (day 18). Importantly, EC-70124 treatment inhibited/delayed AML cell growth for ~24 days after withdrawal of treatment, but disease reoccurrence was eventually observed after this timepoint (Figure 3e). These data reveal that orally dosed EC-70124 inhibits FLT3-ITD^MUT^ AML cell growth in vivo, but withdrawal of the inhibitor results in leukemia relapse, suggesting that outgrowth of EC-70124 resistant cells.

We next performed targeted NGSeq on ex vivo FACS-sorted AML cells retrieved from PDXs treated (n = 2) or untreated with EC-70124 (n = 2) at the endpoint in order to genetically characterize the EC-70124 resistant outgrowing cells observed at relapse. The variant allelic frequency (VAF) of FLT3-ITD after EC-70124 or vehicle treatment in vivo remained almost identical to that at disease presentation (45.47 vs. 46.27), indicating that a FLT3-IDT^MUT^ clone remains the major clone responsible for disease reoccurrence after EC-70124 discontinuation. Interestingly, only one mutation in *EZH2* (VAFs: 6%), a gene recently involved in AML pathogenesis, was specifically found in EC-7012-treated mice (Table 2). These data indicate that AML reoccurrence after suspension of EC-70124 treatment is driven by the major FLT3-ITD^MUT^ AML clone. The biological relevance of the EZH2 mutation in 12% of the FLT3-ITD AML cells requires further prospective experimental work.

## 4. Discussion

AML is the most common hematological malignancy in adults, and the presence of FLT3-ITD^MUT^ confers poor outcome in terms of overall and relapse-free survival in AML patients [41,42]. Indeed, recent results suggest that FLT3-ITD^MUT^ may function as an important oncogenic hit accelerating disease progression [43]. Accordingly, few TKIs disrupting the oncogenic signaling initiated by FLT3 are already FDA-approved (midostaurin and gilteritinib), and several others are under development, offering promising therapeutic strategies for patients with FLT3-ITD^MUT^ AML [44]. In line with this aim, we here evaluated the efficacy of EC-70124, a selective inhibitor from the same chemical space as midostaurin, on FLT3-ITD^MUT^ AML cells. We confirm in vitro the cytotoxic effect of EC-70124 on primary FLT3-ITD^MUT^ AML patient cells and cell lines, revealing an even greater toxicity than midostaurin, which is in line with recent data by Puentes-Moncada et al. [21]. These authors have previously shown that EC-70124 exerts a higher inhibitory activity for FLT3-ITD^MUT^ than midostaurin using kinase affinity assays, pointing to EC-70124 as a novel therapeutic for AML.

EC-70124 is a potent multi-kinase inhibitor that can inhibit a broad spectrum of kinases including FLT3, JAK, SYK and PIM [18,19,20,21,22,23]. FLT3-ITD induces several downstream pro-survival pathways including MAPK, PI3K/Akt and STAT5, triggering proliferation and suppressing apoptosis [45]. EC-70124 treatment increased apoptosis and induced cell cycle arrest in FLT3-ITD^MUT^ cell lines, and these effects were mediated through inhibition of FLT3 signaling. We found that EC-70124 inhibited p-AKT, p-ERK and pSTAT5, a master target in constitutively activated FLT3, and also inhibited the phosphorylation of S6 and BAD, which are downstream components of MAPK or PI3K pathways. This indicates a specific inhibitory function of EC-70124 on FLT3 signaling.

Of note, it has been reported that EC-70124 has more affinity for FLT3 mutants in AML than both wild-type FLT3 and other kinases involved in the development of healthy hematopoietic cells, such as KIT, suggesting safety and specificity for FLT3-ITD^MUT^ [21,46]. Indeed, EC-70124 had no cytotoxic effects on FLT3-ITD^WT^ cell lines, and barely (not significant) impacts FLT3-ITD^WT^ AML patient cells, indicating a high specificity for FLT3-ITD^MUT^ versus FLT3-ITD^WT^ AML cells. This affinity may confer an advantage for treatment with FLT3 inhibitors, as it may uncouple specificity and potency from toxicity caused by off-target activity, the main limitation of first-generation inhibitors [47]. Very importantly, EC-70124 spares healthy HSPCs and mature blood lymphoid and myeloid cells, underlying its safety and non-myeloablative profile on normal hematopoiesis.

Finally, in vivo analysis revealed that EC-70124 has antitumor activity and suppresses disease progression in both MOLM-13 and FT3-ITD^MUT^ PDX models. However, after treatment withdrawal, mice transplanted with FT3-ITD^MUT^ PDX relapsed, with an action window of ~24 days, suggesting the eventual outgrowth of EC-70124-resistant AML cells. These results are in line with previous studies demonstrating that FT3-ITD^MUT^ is subclonal and absent in the most immature AML-initiating/preleukemic cells, reinforcing that therapeutic targeting of subclonal mutations such as FLT3-ITD^MUT^ might only have a transient clinical benefit in debulking the leukemia, but is unlikely to be curative, since it will not target the roots of the disease [48]. In fact, the VAF of FLT3-ITD after EC-70124 treatment in vivo remained almost identical to that at disease presentation, indicating that a FLT3-IDT^MUT^ clone remains the major clone responsible for disease reoccurrence after EC-70124 discontinuation. Emergence of resistance poses a significant challenge in TKI-based treatments [44]. FLT3-TKD point mutations in the FLT3 drug binding site have been previously associated with resistance to FLT3 inhibitors [11]; however, we failed to detect such mutations in the FLT3 gene in EC-70124-resistant AML cells. Acquisition of new mutations in other genes and activation of alternative signaling pathways represent a common but complex mechanism underlying resistance to FLT3 inhibitors [44,49]. Interestingly, only one mutation in *EZH2* (VAFs: 6%), a gene recently involved in AML pathogenesis [50,51], was specifically found in EC-7012-treated mice. The biological relevance of the EZH2 mutation in 12% of the FLT3-ITD AML cells requires further prospective experimental work. In sum, our results suggest that wise treatment schemes should be developed to administer TKIs as co-adjuvant treatment with other therapeutic strategies and/or allogeneic HSC transplantation aimed to target the AML-initiating cells and suppress disease progression.

## 5. Conclusions

Here, we show that EC-70124, a multi-kinase inhibitor, has a potent and selective cytotoxic effect on FLT3-ITD^MUT^ cells in vitro and in vivo, delaying disease progression and sparing FLT3-ITD^WT^ cells and healthy mature hematopoietic cells and HSPCs. Collectively, our results support EC-70124 as a novel and safe TKI in FLT3-ITD^MUT^ AML.

## Figures and Tables

**Figure 1 cancers-14-01593-f001:**
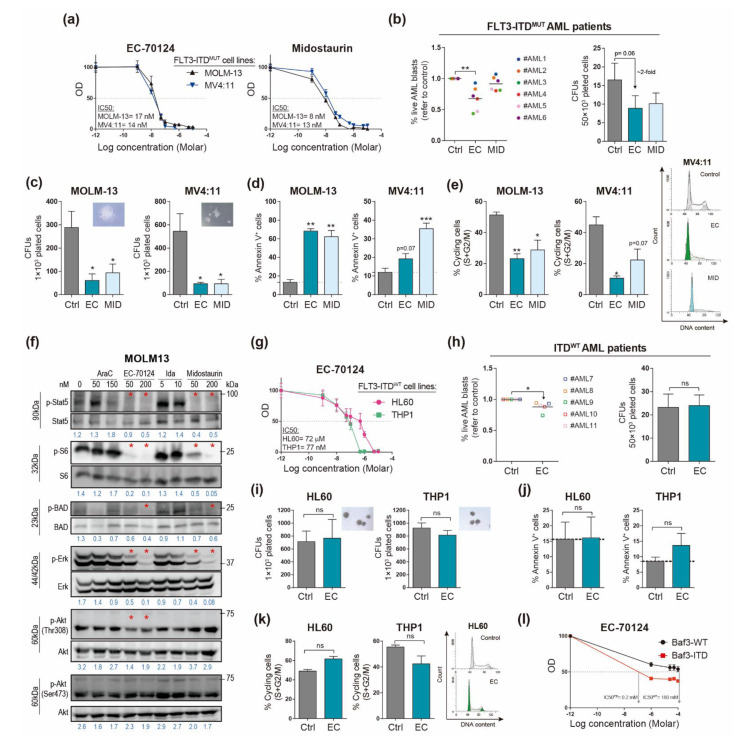
EC-70124 exerts profound cytotoxic effects in FLT3-ITD^MUT^ AML cells. (**a**) Cytotoxicity dose–response curves of midostaurin (right panel) and EC-70124 (left panel) on FLT3-ITD^MUT^ AML cell lines. (**b**) Effect of EC-70124 (EC) and midostaurin (MID) on cell viability (left panel) and CFU formation (right panel) in primary FLT3-ITD^MUT^ AML patient cells. Viability data are relative to the untreated group (Ctrl). Patient samples are color-codified, and black lines represent the mean. (**c**–**e**) Effect of EC-70124 on CFU formation (**c**), apoptosis (**d**) and cell cycle (**e**) in FLT3-ITD^MUT^ AML cell lines. Representative CFUs are shown as inset in C. Representative cell cycle FACS analysis is shown E (right panel). (**f**) Representative Western blots showing the effect of EC-70124 and midostaurin on the main targets related to FLT3 signaling (Appendix A). Lysates from cells treated with cytarabine (AraC) and idarubicin (Ida) were included as controls. Red asterisks depict the specific effect of EC-70124 or midostaurin on the indicated phosphorylation. Blue numbers depict the ratio phosphorylated/total protein for each band. (**g**) Cytotoxicity dose–response curves of EC-70124 on FLT3-ITD^WT^ AML cell lines. (**h**) Effect of EC-70124 on cell viability cell (left panel) and CFU formation (right panel) in primary FLT3-ITD^WT^ AML patient samples. Viability data are relative to Ctrl group. Patient samples are color-codified, and black lines represent the mean. (**i**–**k**) Effect of EC-70124 on CFU formation (**i**), apoptosis (**j**) and cell cycle (**k**) in FLT3-ITD^WT^ AML cell lines (HL60 and THP1). Representative CFUs are shown as inset in (**i**). Representative cell cycle FACS analysis is shown (**k**) (right panel). (**l**) Cytotoxicity dose–response curves of EC-70124 on Baf3-WT and BaF-ITD cells (n = 3). Data represent mean ± SEM. * *p* < 0.05; ** *p* < 0.01; *** *p* < 0.001; ns, not significant (Student’s *t* test). Abbreviations: EC, EC-70124; MID, midostaurin.

**Figure 2 cancers-14-01593-f002:**
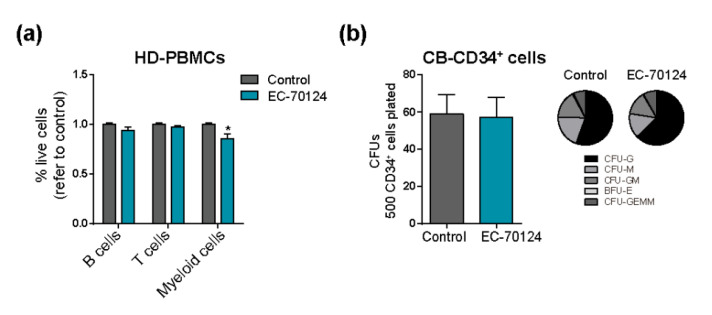
EC-70124 has no cytotoxic effects on healthy hematopoietic (stem/progenitor) cells. (**a**) Cytotoxic effect of EC-70124 on HD-PBMCs (B, T and myeloid cells). (**b**) Number of CFUs from CB-CD34+ cells after treatment with EC-70124. Right panel, scoring of the colony types. * *p* < 0.05 (Student’s *t* test).

**Figure 3 cancers-14-01593-f003:**
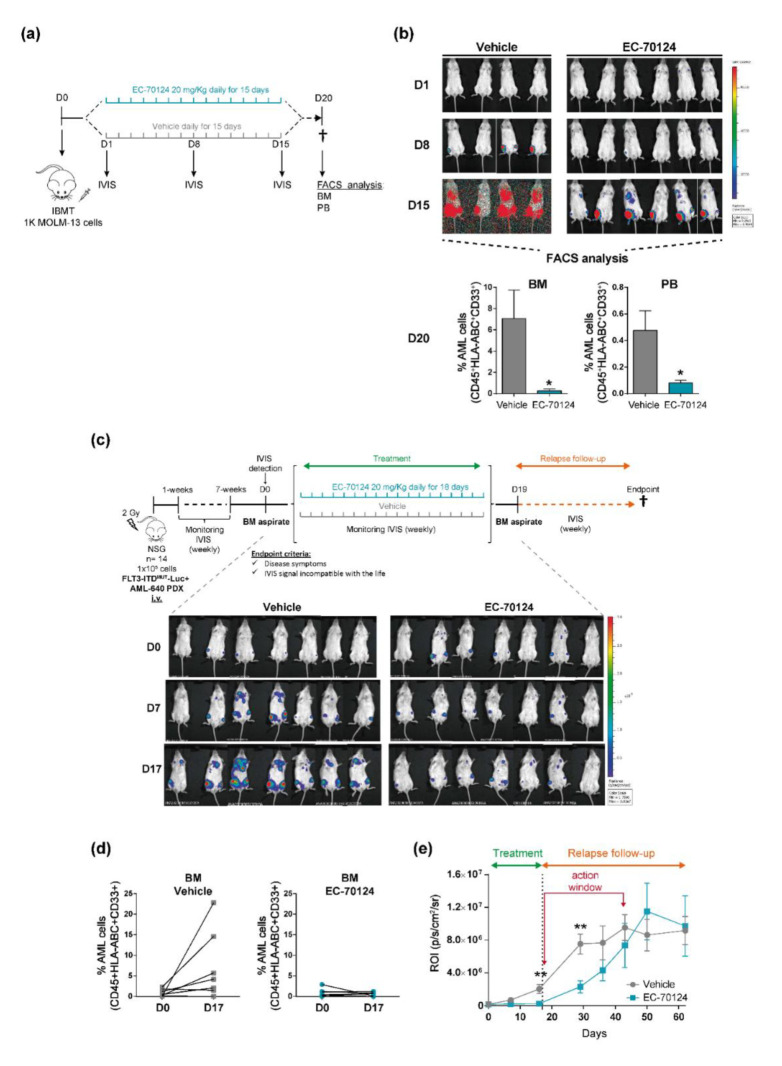
Oral administration of EC-70124 delays the growth of FLT3-ITD^MUT^ AML cells in NSG mice. (**a**) Experimental design for the MOLM-13 leukemic model. (**b**) Upper panel, IVIS imaging of tumor burden monitored by bioluminescence at the indicated time points in the MOLM-13 leukemic model. Bottom panel, percentage of MOLM-13 cells at the endpoint in the BM and PB of vehicle- and EC-70124-treated mice. (**c**) Upper panel, experimental design for the FLT3-ITD^MUT^Luc+ AML-640 PDX model. Bottom panel, IVIS imaging of tumor burden by bioluminescence at the indicated time points in the Luc-FLT3-ITD^MUT^ PDX model. (**d**) Percentage of FLT3-ITD^MUT^ AML blasts in BM at the beginning (D0) and the end (D17) of the treatment in the FLT3-ITD^MUT^ AML PDX model. Each point depicts a mouse. (**e**) Total radiance quantification at the indicated time points. Ec-70124 was stopped on day 17. * *p* < 0.05; ** *p* < 0.01 (Multiple *t* test).

**Table 1 cancers-14-01593-t001:** Biological and cytogenetic-molecular characteristics of patients with AML used in this study.

Patient ID	Diagnostic	Cytogenetics	Molecular	Blasts (%)	Age (y)	Sex
#AML1	AML-M4	46,XX, + 8	NPM1^MUT^, FLT3-ITD	70	63	F
#AML2	AML-M4	46,XY [20]	FLT3-ITD, DNMT3, NRAS	n.a.	64	M
#AML3	AML-M5	n.p	NPM1^MUT^, FLT3-ITD	>90	77	M
#AML4	AML	46,XX,del(5)(q31) [2]/45,XXdel5(q31),-7 [7]	FLT3-ITD	60	76	F
#AML5	AML-M4	46, XY	NPM1^MUT^, FLT3-ITD	88	48	M
#AML6	AML-M4	n.p	FLT3-ITD	88	16	M
#AML7	AML-M2	45,XX,-7 [9]/46,XX [13]	-	68	33	F
#AML8	AML	n.p	NPM1^MUT^	>90	86	F
#AML9	Relapse AML	46,XX [20]	-	40	52	F
#AML10	AML-M4	48,XX, + 8, + 13,inv16(p13;p22) [4]/ 47,XX, + 8inv16(p13;p22) [4]	CBF/MYH11	90	44	F
#AML11	AML-M1	n.p	-	90	81	M

Abbreviations: y, years; M, male; F, female; n.p, not performed due to absence of metaphases; -, no mutations found in TP53, FLT3, NPM1, cEBPa, WT and IDH1; n.a. not analyzed.

**Table 2 cancers-14-01593-t002:** Specific mutations (and their VAF) commonly found in myeloid genes in AML patients analyzed by targeted NGS in ex vivo FACS-sorted AML cells retrieved from PDXs treated (n = 2 mice) or untreated with EC-70124 (n = 2 mice).

		Vehicle		EC70124	
**Gene**	cHgvs	Mouse #1 VAF (%)	Mouse #2 VAF (%)	Mouse #1 VAF (%)	Mouse #2 VAF (%)
**DNMT3A**	c.2173 + 1G > A	44.73	48.90	44.54	47.86
**DNMT3A**	c.1937-8T > A	49.32	47.48	45.09	47.29
**IDH1**	c.395G > A	47.93	46.90	44.94	49.01
**NPM1**	c.860_863dup	32.11	32.27	36.32	34.47
**FLT3**	c.1774_1833dup	44.79	46.15	48.31	44.22
**EZH2**	c.179G > A	-	-	9.65	2.47

Abbreviations: *VAF*: variant allele frequency; -: no pathogenic variants found; black and blue numbers: depict mutations in myeloid genes commonly found in AML in both vehicle- and EC-70124-treated mice (black) or exclusively in EC-70124-treated mice (blue).

## Data Availability

The data presented in this study are available on request from the corresponding author.

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
