# Peer review of "The Multi-Kinase Inhibitor EC-70124 Is a Promising Candidate for the Treatment of FLT3-ITD-Positive Acute Myeloid Leukemia"

_cancers, 2022, doi:10.3390/cancers14061593_

Round 1

Reviewer 1 Report

Article is ready to publish

Reviewer 2 Report

The authors have now answered my questions and responded appropriate to my comments. I recommend to accept the manuscript.

This manuscript is a resubmission of an earlier submission. The following is a list of the peer review reports and author responses from that submission.

Round 1

Reviewer 1 Report

This is a professionally written high quality work with elevated translational potential. Experiments are properly designed and results sustain the conclusions drawn.

Some aspects could be discussed more critically:

  • In MV4:11 cells EC-70124 compared to midostaurin features a minor pro-apoptotic effect and a more pronounced cell cycle arrest
  • 1e righ panel is nearly unreadable, maybe translate to supp. materials?
  • In Fig.1f indicate the position of MW markers not the expected size of the protein. I do not agree no decrease of pERK. Data must be quantified and analyzed with replicates.
  • The problem of resistance is not addressed nor discussed, do you expect that mutations promoting resistance to other FLT3 inhibitors will also arise with EC-70124?

Reviewer 2 Report

In this study by Lopez-Millan et al. the effect of the inhibitor EC-70124 on acute myeloid leukemia cells with a FLT3-ITD was tested, and compared to the effect of midostaurin. In vitro, EC-70124 exerted a robust and specific activity against FLT3-ITDmut AML primary AML cells and cell lines with respect to cytotoxicity, CFU capacity, apoptosis and cell cycle while sparing healthy hematopoietic (stem/progenitor) cells. In vivo, EC-70124 was also studied for its efficiency against Molm13 and primary AML cells in xenograft models.

This is a very nice study with an inhibitor, EC-70124, that shows a high potential in the eradication of FLT3-ITD-positive AML cells. However, all (or most of) the results on the effect of EC-70124 and midostaurin on AML cell lines that are depicted in Figure 1 are already published in Puente-Moncada et al. These results should be removed as much as possible from Figure 1, and the data with primary AML cells should be expanded. For example, the authors should perform experiments with the inhibitor and primary AML cells and measure apoptosis of the various AML cell fractions (blasts, CD34+ cells). Moreover, the induction of differentiation after incubation with the inhibitor should be measured. There is one sample in the collection of wild type samples that is a relapse sample. This one should be removed as those cells probably respond less to every therapeutic, and the collection of wild-type samples should be expanded till at least six samples (similar as the number of FLT3-ITD positive samples).

The experiments presented in Figure 1F are already shown in the paper of Puente-Moncada, showing that EC-70124 is reducing p-Stat5 and affecting P-Bad and S6. The authors should show this with primary AML cells.

In the study by Puente-Moncada et al. in Molecular Cancer Therapeutics on EC-70124 it seems to me that EC-70124 is also effective, although with higher IC50 values, against cell lines with wild-type FLT3 (such as Molm16 and KG1). In this paper the authors selected that non-responsive cell lines HL60 and THP1 but as Puente-Mocanda already showed it is not true for all wild-type cell lines. How are the authors explaining this?

As primary AML blasts in culture are hardly proliferating it might be that EC-70124 is able to target quiescent blasts better than midostaurin (Figure 1B). The effect of EC-70124 on the cell cycle in AML cell lines was already shown in the paper of Puente-Moncada et al. The right figure in Figure 1E and Figure 1K are unreadable. To measure colony forming capacity of cell lines is not very informative.

The results on the effect of EC-70124 on primary AML cells and on the BAF cells transduced with FL3-ITD are very nice and shows the specificity of the EC-70124 for cell with a FLT3-ITD. However, the experiment with the BAF-FLT3-ITD cells should be performed in triplicate, and therefore repeated as it looks from Figure 1I that this has been done one time. Moreover, the overexpression of FLT3-ITD (and enhanced activity of FLT3) in the BAF cells should be shown.  

Figure 1E: withdrawal of the inhibitor results in rapid growing of the leukemia (even harder than the  control cells as the leukemia is the same size at 60 days) and suggests that the treatment has to be continued much longer. The authors should measure the presence of the FLT3-ITD in the leukemia cells at the various time points after stopping the treatment, to investigate whether another clone without the mutation grows out to form the leukemia relapse.

Is the inhibitor also effective in AML with a FLT3-TKD mutation?